# Organ Crosstalk in Acute Kidney Injury: Evidence and Mechanisms

**DOI:** 10.3390/jcm11226637

**Published:** 2022-11-09

**Authors:** Xiaolong Li, Feifei Yuan, Lili Zhou

**Affiliations:** State Key Laboratory of Organ Failure Research, National Clinical Research Center of Kidney Disease, Division of Nephrology, Nanfang Hospital, Southern Medical University, Guangzhou 510515, China

**Keywords:** AKI, organ crosstalk, clinical and laboratory evidence, molecular mechanisms

## Abstract

Acute kidney injury (AKI) is becoming a public health problem worldwide. AKI is usually considered a complication of lung, heart, liver, gut, and brain disease, but recent findings have supported that injured kidney can also cause dysfunction of other organs, suggesting organ crosstalk existence in AKI. However, the organ crosstalk in AKI and the underlying mechanisms have not been broadly reviewed or fully investigated. In this review, we summarize recent clinical and laboratory findings of organ crosstalk in AKI and highlight the related molecular mechanisms. Moreover, their crosstalk involves inflammatory and immune responses, hemodynamic change, fluid homeostasis, hormone secretion, nerve reflex regulation, uremic toxin, and oxidative stress. Our review provides important clues for the intervention for AKI and investigates important therapeutic potential from a new perspective.

## 1. Introduction

Acute kidney injury (AKI) is a severe condition and is associated with a worsened prognosis in critically ill patients. Clinically, AKI is determined based on elevated serum creatinine levels (a marker of the kidney excretory function) and reduced urinary output (a quantitative marker of urine production) within 7 days. According to the KDIGO definition, AKI can be classified into three stages (1–3) relying on a list of functional criteria (Table 1) [1,2]. However, these criteria do not help to diagnose the early stage of AKI (so-called “subclinical AKI”), meaning that the best treatment time for AKI is easily missed. Thus, before the KDIGO criteria for AKI are met, increasing biomarkers that help to indicate kidney injury are discovered by persons. Currently, only neutrophil gelatinase-associated lipocalin (NGAL), tissue inhibitor of metalloproteinase 2 (TIMP-2), and insulin-like growth factor-binding protein 7 (IGFBP7) are available for clinical determination of subclinical AKI [3,4,5,6]. Notably, AKI is one condition within a group of acute kidney disease and disorders (AKD) which can occur without ever meeting the criterion of AKI and can continue to exist when kidney structure damage is persistent. When AKD lasts longer than 3 months, it is referred to as chronic kidney disease (CKD) (Table 1) [7].

AKI can be induced by a broad range of causative factors depending on different clinical settings, such as sepsis, hemorrhage, low output after cardiac surgery, and the application of chemotherapy drugs. These factors contribute to the high incidence of AKI worldwide. According to a recent meta-analysis of 154 studies, the incidence of AKI was 8.3% in ambulatory patients, and as high as 20–31.7% in patients at various levels of in-hospital care [8]; moreover, the ratio increases to 57% in critically ill patients [9]. The symptoms of kidney failure, including uremic toxin accumulation, metabolic acidosis, electrolyte imbalances, and fluid overload, are the typically well-known outcomes of AKI that contribute to high mortality [10]. However, a significant proportion of AKI-associated mortality cannot be explained simply by the loss of kidney function or by complications occurring during AKI treatment. Multiple organ dysfunction caused by AKI is a particularly important consequence in critically ill patients [11]. According to a multicenter of AKI cohort study, cardiac failure was found in 60% of patients with AKI in the intensive care unit (ICU) [12]. A meta-analysis including 254,150 adults (55,150 with AKI) also concluded that AKI increased the risk of subsequent heart failure by 58%, myocardial infarction by 40%, and stroke by 15% [13]. Moreover, the average mortality rate of AKI was 23% but reached 45–60% in those cases accompanied with other organic dysfunction, such as liver and heart failure [8,14,15]. These data suggest organ crosstalk can occur in AKI and plays a critical role in patients’ prognosis.

Accumulating evidence indicates that the communication between injured kidneys and other organs, including the heart, lung, liver, brain, and gut, is mediated by a range of inflammatory cytokines and immune cells [16]. However, due to the limited genetic diversity of animals and the lack of suitable animal models which mimic the organ crosstalk in AKI patients [17], the molecular mechanisms of AKI-induced multiorgan failure are still unclear. In this review, we summarize the updated findings of crosstalk between injured kidneys and other organs from clinical and laboratory studies. Furthermore, we summarize the underlying molecular mechanisms of organ crosstalk in AKI (Figure 1).

## 2. Kidney–Lung Crosstalk

In the context of the global coronavirus disease 2019 (COVID-19) pandemic, the effects and mechanisms of lung–kidney communication are of great interest to the scientific community and commendable progress has been made [18]. As a respiratory organ, the functions of the lung are not limited to gas exchange, but also include immunomodulatory [19], hematopoietic [20], secretory [21], and metabolic functions, which play a role in kidney–lung crosstalk under physiological conditions. For example, one of the secreted factors from pulmonary vascular endothelial cells (PVEC), prostacyclin (PGI_2_), can directly or indirectly act on the kidney through a variety of signaling pathways to regulate renal blood flow (RBF), glomerular filtration rate (GFR), glomerular and tubular development, renin secretion and other pathophysiological processes [22], suggesting the existence of kidney–lung crosstalk.

In addition, the lung and kidneys work together to ensure the stability of blood pressure through the renin–angiotensin–aldosterone system (RAAS) [23] and kallikrein–kinin system [24], and the fluid acid-base equilibrium by regulating bicarbonate and carbon dioxide concentration [25]. Moreover, the specific structural similarities of the basement membrane and its compositions in the kidney and lung suggest the possibility of kidney–lung interaction, which partly explains the pathogenesis of Goodpasture’s syndrome [26]. Interestingly, the alveolar and renal tubular epithelium share common characteristics at the polarizing sites (apical and basolateral) in the localization and distribution of transporters, ion channels, and tight junctional complexes, which also provide the structural basis for lung–kidney communication [27].

### 2.1. Clinical Evidence

Several clinical epidemiological studies have shown that several types of renal dysfunction, including hematuria, proteinuria and AKI, are the most common complication of acute respiratory distress syndrome (ARDS). An observational study of 8029 critically ill patients (including 1879 patients with ARDS) found that the incidence rate of AKI in patients with ARDS was higher than in those without ARSD (44.3% versus 27.4%, *p* < 0.001), suggesting that ARSD is an independent risk factor for AKI [28]. Moreover, a meta-analysis involving 31 studies and 10,333 patients revealed a strong association between the administration of invasive mechanical ventilation (IMV) and the risk of developing AKI, the pooled odds ratio (OR) of which was 3.58 (95% CI 1.85 to 6.92; *p* < 0.001) [29]. In addition to acute lung injury (ALI), AKI also occurs in chronic lung diseases, such as chronic obstructive pulmonary disease (COPD), and its incidence rate is 128/per 100,000 person-years, but the mortality rate is as high as 521/1000 person-years [30].

Conversely, the adverse effects of AKI on the lung are mainly reflected in the increasing frequency and duration of IMV and the susceptibility to respiratory complications and respiratory failure [31,32,33,34]. One study found that patients with AKI are more than twice as likely to develop respiratory failure and nearly three times as likely to die compared with those without AKI [35]. Additionally, a clinical study found that patients with AKI require a longer IMV duration and weaning from mechanical ventilation, suggesting that recovery from lung dysfunction appears to be affected by AKI [31].

### 2.2. Laboratory Evidence

Besides the clinical research, kidney–lung communication has also been observed in various animal models and it contributes to the development of lung and kidney disease individually or synergistically. In the ischemia-reperfusion (IR)-induced AKI mouse model, Hassoun et al., revealed that 66 apoptosis-related genes and 29 inflammation-related genes were activated in the lung; moreover, the activated apoptotic genes were mainly located in pulmonary endothelial cells, indicating that it may be an important mechanism of ALI [36]. Similarly, Rabb et al. also found that bilateral IR and bilateral nephrectomy models were more prone to lung injury and pulmonary dysfunction, whereas unilateral IR models were not [37]. Likewise, in a *P. aeruginosa*-induced pneumonia model, researchers revealed strong evidence of tubular injuries, such as elevated plasma creatinine, cystatin C, urinary NGAL, and the pathological phenomenon of brush-border collapse and mild tubular cell apoptosis [38]. Similar results were observed in ARDS rat models induced by intratracheal lipopolysaccharide installation [39], and in ventilator-induced lung injury mice [40].

### 2.3. Mechanism

Under pathological conditions, the mechanisms of kidney–lung crosstalk involve an inflammatory response (e.g., imbalanced immune response and increased inflammatory mediators, etc.) and a fluid homeostasis imbalance caused by kidney or lung injury (e.g., fluid overload, uremic toxin retention, hypoxia, and hypercapnia, etc.) [34] (Figure 2).

#### 2.3.1. Inflammation and Immune Imbalance

Systemic inflammatory response syndrome (SIRS) characterized by increased levels of circulating cytokines and chemokines, including IL-1β, IL-6, IL-8, MCP-1, and TNF-α, was observed in AKI or ALI patients and animal models, which may cause lung and kidney inflammation, cell apoptosis, increases in endothelial barrier permeability, oxidative stress, and aggravation of pulmonary edema [33,34,41]. Among these pro-inflammatory factors, IL-6 was considered to be a more critical mediator of kidney–lung crosstalk, and IL-6 deficiency protected against AKI-induced lung injury via reductions in the level of IL-8 in lung and serum, thereby diminishing pneumonia and capillary leakage [42]. However, only circulating IL-6 seemed to have these deleterious effects, whereas direct intratracheal injection of IL-6 showed an anti-inflammatory effect, further suggesting that IL-6 is involved in kidney–lung communication [43]. Furthermore, immune cells also play a key role in kidney–lung communication. Lung histopathological analysis following AKI showed that renal injury facilitates a series of lung tissue immune responses including monocytes, neutrophils, and CD8^+^ T cell invasion [44,45,46].

#### 2.3.2. Fluid Overload and Uremic Toxin

There is sufficient evidence to demonstrate that pulmonary edema after AKI is not only associated with fluid overload caused by renal failure, but also with reduced pulmonary fluid clearance capacity due to the down-regulation of pulmonary Na, K-ATPase, epithelial sodium channels (ENaC), and aquaporin-5 expression [37,47]. Several studies have shown that indoxyl sulfate, a small protein-bound uremic toxin, exacerbated pulmonary edema and inflammation by downregulating water clearance proteins and inducing IL-6 expression [37,48]. Additionally, during AKI, the retention of uremic toxins negatively affects lung function, as indicated by vital capacity (VC) and maximal breathing capacity (MBC) [49].

#### 2.3.3. Hypoxia and Hypercapnia

Furthermore, given that the kidneys are extremely vulnerable to hypoxia, even short-term hypoxemia can adversely affect renal function, and lung injury-induced hypoxemia and hypercapnia inevitably influence renal function. Hypoxemia and hypercapnia have been demonstrated to, directly or indirectly, decrease renal blood flow, resulting in renal dysfunction [50,51,52]. On the one hand, hypoxemia and hypercapnia directly stimulate renal vasoconstriction by promoting the release of norepinephrine [53]; on the other hand, sustained high plasma CO_2_ can trigger systemic vasodilation, which then activates the RAAS in a feedback manner and indirectly reduces renal blood flow to induce AKI [50].

#### 2.3.4. Novel Mediators

Furthermore, increasing numbers of novel mediators have been identified in kidney–lung interactions (Figure 2). Recently, Hepokoski et al. first discovered that metabolic change and mitochondrial damage associated molecular patterns (mt-DAMPs), such as the release of mitochondrial DNA (mt-DNA), play a critical role in kidney–lung interaction. The mechanism of mt-DAMPs molecules from the injured kidney in aggravating the pulmonary injury and mitochondrial dysfunction was verified by performing metabolomic analysis of the kidney, lung, and plasm [54]. Moreover, Khamissi et al. found that the kidney injury-related molecule osteopontin (OPN) was a novel agent for AKI–ALI communication [55]. Furthermore, αKlotho, primarily produced and released by the kidney, appears to play a protective role in renal–lung communication. The deficiency of αKlotho aggravates lung injury after AKI, exogenous supplementation of αKlotho can improve lung function by increasing antioxidant capacity [56].

## 3. Kidney–Heart Crosstalk

As the “engine” of human beings, the heart possesses a strong blood pumping capacity and constantly provides sufficient blood to the body’s tissues and organs, relying on rhythmic contraction and relaxation. The normal cardiac output of a healthy adult male at rest is about 5 L/min, and about 20% of this output is received by the kidney, indicating a tight functional connection between the heart and kidney [57]. The relationship of communication between the kidney and heart in regulating blood volume and pressure has been investigated for well over a century. Under physiological conditions, the function of the cardiovascular system is largely regulated by the kidney through the synthesis and release of renin, erythropoietin, endothelin, prostaglandin, and other active substances [58]. Correspondingly, the kidney functions of water and sodium excretion are regulated by myocardial cells through the release of vasopressin (VAP) [59], atrial natriuretic peptide (ANP) [60], and brain natriuretic peptide (BNP) [61].

As early as 1836, Robert Bright first observed structural changes in the hearts of patients with advanced kidney disease [62]. Since then, significant advances have been made in the clinical epidemiology of cardio–renal communication, and the term “cardiorenal syndrome” (CRS) was coined to explain this relationship. According to the different pathological features of CRS, the Acute Dialysis Quality Initiative (ADQI) outlined a new classification of CRS with five subtypes based on sequential organ involvement and disease acuity, respectively (Table 2) [63,64]. Among these subtypes, type 1 CRS describes an AKI that occurs following acute worsening of cardiac function, and type 3 CRS describes a heart injury or dysfunction due to AKI [65]; both these types are discussed in this review.

### 3.1. Clinical Evidence

AKI which occurs following acute cardiac events, such as acute decompensated heart failure (ADHF), acute coronary syndrome (ACS), cardiogenic shock, and cardiac surgery, is classified as type 1 CRS [66]. A large clinical study involving more than 50,000 adult patients found that approximately 39% of patients developed AKI after a range of major surgeries during the index hospitalization [67]. Data from several studies have shown that approximately 20–40% of patients with ADFP develop kidney injury [68,69]. Furthermore, AKI in the setting of acute heart failure is associated with longer hospitalization, and higher readmissions [70], and the mortality at postoperative day 90 of patients with AKI showed a 1.48-fold increase when compared with those who did not develop AKI [67].

Conversely, type 3 CRS is clinically used to define cardiac injury caused by various AKI. As early as 2003, the American Heart Association defined both proteinuria and a decline in glomerular filtration rate (GFR) as independent risk factors for the development of cardiovascular disease [71]; however, there are uncertainties in the epidemiology of type 3 CRS. Currently, epidemiological research suggests that AKI predisposes patients to cardiac events, such as acute heart failure and acute arrhythmias, via multiple direct or indirect pathways, including volume overload, metabolic acidosis, and electrolyte disorders (including but not limited to hyperkalemia and hyperphosphatemia) [72]. More in-depth studies are needed to elucidate the risk factors of AKI-induced CRS.

### 3.2. Laboratory Evidence

In addition, the heart–kidney pathophysiological interaction also has been detected in various animal models. In the rat coronary artery ligation-induced myocardial infarction (MI) model, Lekawanvijit et al. found increased IL-6, TGF-β, Kim-1 expression, and macrophage infiltration in the kidney within 1 week [73]. Moreover, in a cardiac arrest and cardiopulmonary resuscitation rat model, the levels of serum creatinine and BUN rose at 24 h, and histological changes in the kidney such as glomerular collapse, renal tubular cell swelling, and inflammatory cell infiltration were also observed [74].

On the other hand, in a rat model of ischemia-reperfusion injury (IRI), the apoptotic myocardial cells were increased in the heart, indicating that AKI results in the alterations observed in the heart which is important in the morbidity and mortality observed clinically [75].

### 3.3. Mechanism

#### 3.3.1. Hemodynamics

Kidney–heart communication is most directly reflected in hemodynamic changes (Figure 3). The traditional explanation for the regulation and mechanism of CRS mainly focuses on the low-flow theory, which in this context, is characterized by renal hypoperfusion because of a low cardiac output and ejection fraction [76]. Inadequate renal afferent blood flow can trigger neurohormonal mechanisms which further lead to renal afferent arteriole vasoconstriction and progressive deterioration of renal function [77]. Therefore, improving renal blood flow by using vasodilators has been recommended for the management of CRS patients with AKI. However, it should be noted that not all patients with heart failure suffer from reduced cardiac output, and there are still a significant proportion of patients who can maintain normal blood pressure or systolic function [70]. Organ congestion and central venous pressure also play important roles in driving kidney dysfunction in patients with ADHF [70,78]. Thus, blood hypoperfusion and congestion of the heart and kidney, individually or simultaneously, contribute to the morbidity of CRS and AKI.

#### 3.3.2. Neurohormonal Hyperactivity

The kidney and heart are linked through neurohumoral mechanisms involving the sympathetic nervous system (SNS), RAAS, and AVP (Figure 3). In the early stages of heart failure, the activation of SNS, RAAS, and VAP is increased, which results in renal vasoconstriction, sodium retention, and fluid overload. Moreover, the abnormal SNS activation and high level of angiotensin-II contribute to tubular cell hypertrophy, apoptosis, necrosis, ROS production, and inflammation occurring in the kidney, eventually triggering AKI [79]. Angiotensin-II, on the other hand, elicits cardiotoxicity, which is characterized by increasing cardiac pathological hypertrophy and remodeling, further worsening the prognosis of CRS [80]. Therefore, angiotensin-converting enzyme (ACE) inhibitors and angiotensin-receptor blockers (ARBs) can improve AKI and CRS via multiple pathways.

#### 3.3.3. Inflammation and Immune Imbalance

Persistent inflammatory stress is associated with immune imbalance, both of which play a critical role in the pathological communication between the kidney and heart (Figure 3). Systemic inflammation, which includes the activation and release of large quantities of inflammatory factors, such as TNF-α, IL-1β, and IL-6, has been generally identified as a common pathological characteristic of both acute and chronic heart failure [81]. In addition, in patients with heart failure, hemodynamic changes characterized by high central venous pressure and peritoneal pressure mechanically damage renal endothelial cells, followed by intravascular coagulation, platelet activation, and release of pro-inflammatory factors, which directly induce renal injury [82]. Notably, Ronco et al. found lower hemoglobin levels in CRS type 1 patients compared with those without AKI, suggesting that hyperactivated inflammation damages the hematopoietic function and that anemia is a risk factor for exacerbating heart failure [83]. On the other hand, in ischemia-reperfusion and cisplatin-induced AKI models, augmented serum levels of TNF-α, IL-1β, IL-6, and INF-γ were detected, which hurt myocardial function [74]. In AKI, damaged renal tubules and endothelial cells release a mass of chemokines to recruit leukocyte infiltration into heart tissue, which is thought to be a key mechanism in cardiac injury [84]. Hence, anti-inflammation therapy is an effective strategy for AKI-related CRS.

## 4. Kidney–Vascular System Crosstalk

The kidney has a rich vascular network (glomerular capillaries and peritubular capillaries) and abundant blood flow (20% of cardiac output), which provide sufficient oxygen and nutrition for tubules to ensure their physiological function. Clinically, most of the pathogenesis of AKI can be attributed to ischemic injuries, such as kidney transplantation, cardiac surgery, sepsis, and shock-induced AKI [57,85]. Hence, abnormal vascular function potentially promotes the progression of AKI.

The physiological connection between renal tubules and blood vessels helps to regulate renal blood flow. For example, when the renal distal convoluted tubule macula densa senses a change in the concentration of Na^+^ in the lumen, it can send a signal, causing the afferent arteriole contraction or relaxation. This phenomenon is called tubuloglomerular feedback. In addition, active substances such as renin, endothelin, and prostaglandin are released from the glomerular complex to the vessels, which also participate in the regulation of renal blood flow [86].

### 4.1. Clinical Evidence

Epidemiological studies have shown that vascular injury plays a pivotal role in the development of AKI caused by various etiologies. Plasma endothelial injury molecules are strong and independent predictors of AKI incidence and are positively associated with AKI mortality [87,88]. A study of patients with acute myocardial infarction (AMI) found that plasma angiopoietin 2 (Ang-2) levels (6338.28 ± 5862.77 versus 2412.03 ± 1256.58 pg/mL) and thrombomodulin (TM) levels (7.6 ± 2.26 versus 5.34 ± 2.0 ng/mL) were higher in patients with AKI than in those without AKI [88]. Similar results of plasma Ang-2 level were seen in patients with AKI following cardiac surgery [89]. Hence, targeting vascular function may be a new strategy to improve the long-term outcomes of AKI.

### 4.2. Laboratory Evidence

The communication between the kidney and the vasculature is classically manifested in sepsis-induced AKI, where interstitial inflammation and thrombosis are the more predominant pathological changes [90]. In animal models of sepsis-AKI, decreased glomerular blood flow was attributed to the inhibition of endothelial nitric oxide synthase activation in arterioles and glomeruli, whereas decreased cortex peritubular capillary perfusion was associated with epithelial redox stress [87,91]. Moreover, upregulation of adhesion molecules in the endothelium leads to leukocyte recruitment, and a reduction in circulating sphingosine 1-phosphate and the loss of components of the glycocalyx in glomerular endothelial cells results in increasing microvascular permeability, both of which further demonstrate the important role of vascular endothelial system injury in the development of AKI.

### 4.3. Mechanism

#### 4.3.1. Ischemia and Low Blood Flow

Pre-renal ischemia is considered to be the major mechanism of AKI [77]. Thanks to renal autoregulation and tubuloglomerular feedback mechanisms, a mild reduction in circulating blood volume does not significantly affect renal blood flow and GFR [92]. However, a rapid drop in blood pressure, severe hypovolemia, and dehydration can lead to renal hypoperfusion, which subsequently triggers renal baroreceptors, then causes renin release, RAAS activation, renal vasoconstriction, and finally worsens AKI [93]. Besides the effects on the kidneys, the activation of the sympathetic nervous system and RAAS, and the secretion of arginine vasopressin potentially increase systemic vasoconstriction and reduce venous return for blood transfusion, which not only leads to ischemic damage to other organs but also increases the risk of circulatory dysfunction [94].

#### 4.3.2. Fluid Overload and Venous Hypertension

Although ischemia and low-flow theory can partially explain prerenal AKI, it is still subject to several clinical challenges. Many studies have demonstrated that high central and renal venous pressure may play an important role in AKI [95,96]. On the one hand, elevated central venous pressure leads to renal venous hypertension, which directly reduces arteriovenous differential pressure and effective glomerular filtration pressure. On the other hand, the decreased intraglomerular pressure and GFR can trigger compensatory mechanisms and activate the neurohumoral axis, leading to increased water and sodium reabsorption, all of which eventually resulting in oliguria and worsening venous congestion [97].

#### 4.3.3. Endothelial Responses

Endothelial cells are located in the inner layer of blood vessels and are able to mount rapid and specific responses to various injury stresses in AKI [91,98]. The response of endothelial cells to AKI mainly includes three aspects: (1) increased vascular permeability leading to vascular leakage; (2) blood coagulation imbalance leading to vascular embolism; and (3) inflammation and leukocyte recruitment [91,99].

Vascular endothelial growth factor (VEGF) and its receptors (VEGFR) are important regulators of angiogenesis and vascular permeability [100]. In sepsis-AKI mice, studies have shown that increased plasma VEGF levels and decreased local renal VEGFR expression cause decreased glomerular endothelial fenestrae density and increased systemic microvascular permeability, suggesting that VEGF-VEGFR are jointly involved in regulating endothelial injury after AKI [91,101]. In addition, the imbalance of angiopoietin (Ang) family proteins and the endothelial-specific Tie2 receptor (decreased expression of Tie2 and Ang1, and increased ANG2) in patients with AKI after cardiac surgery leads to vascular instability [89]. Pretreatment with Tie2 agonists can improve renal function in sepsis-AKI mice [102]. Furthermore, the down-regulation of sphingosine 1-phosphate (S1P)-S1PR signaling pathway is also involved in endothelial permeability impairment in AKI [103].

Thrombosis may occur and aggravate ischemia and hypoxia injury during AKI. Notably, coagulation abnormalities may occur in all capillaries and show unique heterogeneity. For example, the loss of endothelial protein C receptor (EPCR) may occur in arterioles, and thrombomodulin (TM) is mainly lost in peripheral capillaries, which may partly explain the failure of single anticoagulant agents to significantly improve renal function [91,104,105].

During AKI, multiple adhesion mediators, such as P-selectin, E-selectin, vascular cell adhesion molecular 1 (VCAM1), and intercellular adhesion molecule 1 (ICAM1), are upregulated in the endothelial cells [106,107]. Through these adhesion molecules, neutrophils, lymphocytes, and macrophages are recruited and accumulate in injured lesions, which then secrete inflammatory factors to cause local and systemic inflammatory reactions [108].

## 5. Kidney–Liver Crosstalk

The liver is the largest metabolic organ, whereas the kidney is the major excretory organ. Fatefully, they cooperate to maintain body homeostasis; in other words, they are functionally linked. Hence, the metabolites are the major mediators of kidney–liver communication under physiological conditions. For instance, the energy of the kidney is derived from fatty acids that are mainly produced in the liver, which means that liver dysfunction inevitably affects renal function. Moreover, various toxic or insoluble substances (drugs), and hormones are increased in solubility or inactivated by the liver and then excreted by kidneys [109]. When the kidneys are not able to excrete these metabolites immediately, the liver is under greater metabolic pressure and damage can occur.

### 5.1. Clinical Evidence

Hepatorenal syndrome (HRS) describes a kidney function impairment in patients with severe liver disease, particularly advanced cirrhosis, without substantial alterations in kidney histology. In HRS, there is a notable reduction in GFR and an increase in serum creatinine levels. According to the rate of renal function decline, the HRS is subclassified into two clinical types. Type 1 is defined as a rapid reduction in renal function via a 50% reduction in creatinine clearance in less than two weeks, and in the initial 24 h, below 20 mL/min or an increase in initial serum creatinine to a concentration of at least 2.5 mg/dL. This pattern of HRS is now known as acute kidney injury (AKI)-HRS. Type 2 is the chronic impairment of kidney function, and this pattern of HRS is now known as chronic kidney disease (CKD)-HRS [110,111]. In a 10-year follow-up study of 234 patients with cirrhosis and ascites, the incidence of hepatorenal syndrome was 18% at 1 year and 39% at 5 years; moreover, there was an up to a 40% probability of developing AKI-HRS within 5 years [112]. In a prospective observational study of 547 patients who were admitted for cirrhosis with acute decompensation, a total of 290 patients had AKI (53%). Among those with AKI, 65% had AKI at the time of admission and 35% developed AKI during their hospitalization [113]. Notably, the development of AKI in cirrhosis patients is associated with high mortality, and the 90 days survival of patients with AKI-HRS is usually less than 20% [114]. Early diagnosis and intervention are necessary.

Unlike the prominent clinical manifestations of HRS, there are limited clinical reports explicitly describing the effects of AKI on the liver. The abnormal liver function that appears in patients with critically AKI is associated with high in-hospital mortality [11]. In addition, many clinical observations have shown that AKI does affect the normal drug metabolism of the liver, which may be one of the reasons for the high mortality of AKI complicated with liver dysfunction [11,16].

### 5.2. Laboratory Evidence

Although the changes in renal injury markers (serum creatinine, BUN, Kim-1, and pro-inflammatory factor expression) with time can be observed in a mouse liver cirrhosis model induced by bile-duct ligation (BDL) [115], human HRS, especially AKI-HRS, cannot be completely replicated. Though Shah et al. successfully established an AKI in a rodent model of cirrhosis through treatment with BDL and LPS [116], scientists still face challenges in establishing an AKI-HRS model without direct nephrotoxicity.

However, several experimental studies have shown that AKI promotes oxidative stress, inflammation, apoptosis, and tissue damage in hepatocytes and increases vascular permeability and leukocyte infiltration into the liver [117,118,119,120]. Furthermore, AKI significantly impacts the hepatic function of drug metabolism. For example, the normal liver clearance rate of vancomycin is 40 mL/min, but it was reduced to 15 mL/min after AKI [14]. These results suggest the close interaction between the kidney and liver.

### 5.3. Mechanism

#### 5.3.1. Circulatory Dysfunction

A series of vasodilators and vasoconstrictors play a role in the communication of the liver and kidney in HRS. Under cirrhosis conditions, intrahepatic vascular resistance is elevated and the vasodilators, such as nitric oxide, carbon monoxide, prostaglandins, and endocannabinoids, are increased, resulting in splanchnic arterial vasodilation and effective arterial blood volume (EABV) reduction. Consequently, systemic vasoconstrictor pathways, such as the RAAS, sympathetic nervous system, and arginine vasopressin (AVP), are activated as compensatory mechanisms for increasing the EABV. These mechanisms result in reductions in renal blood flow and the glomerular filtration fraction, eventually, leading to AKI [111,115,121].

#### 5.3.2. Bile Acid and Bilirubin

Bile acid and its related metabolites, cytokines and chemokines, are mediators of kidney–liver crosstalk. Histopathologic studies have shown that 18–75% of patients with AKI-HRS present intratubular bile acid casts which can directly poison tubular cells and elicit tubulointerstitial inflammation, lipid peroxidation, and oxidative stress in the kidney [11,122,123]. In addition, serum bilirubin levels, urinary bilirubin, and urobilinogen are elevated in most patients, and serum bilirubin concentrations above 10 mg/dL are associated with a worse prognosis in patients with AKI-HRS compared with patients with serum bilirubin below 10 mg/dL [124,125]. Thus, changes in bile acid and bilirubin are not negligible points in the management of AKI-HRS patients.

#### 5.3.3. Systemic Inflammation

Systemic inflammation occurs in almost half of the patients with AKI-HRS, independent of the presence of infection [126]. The increase of serum cytokines and chemokines, including IL-6, TNF-α, and MCP-1, promote tubular injury, and neutrophil, macrophages infiltrate into the kidney. Accordingly, patients with AKI-HRS show increased expression of toll-like receptor 4 (TLR4) and caspase-3 in tubular cells, and reducing TLR4 expression markedly elicits the nephroprotective effect [127]. On the other hand, AKI-induced liver injury also occurs due to various cytokines, such as IL-6, and IL-17A. Park et al. suggests that in wild-type mice, treatment with neutralizing antibodies against TNF-α, IL-17A or IL-6 has a protective effect on hepatic and small intestine injury because of ischemic or non-ischemic AKI [119]. However, the mechanism of AKI-induced liver dysfunction of drug metabolism is still unclear. It may be related to the decreased expression of the drug-metabolizing enzyme CYP3A11, and drug transporters, such as MDR1a, MRP2, and OATP3, in the liver and intestine [128]. Nevertheless, efforts are needed in further investigation.

## 6. Kidney–Gut Crosstalk

A healthy gut contains the largest absorptive surface in the body, 70–80% of the body’s immune cells, and at least 1000 genera of bacteria, which together form three barriers including the physical, immune, and biological barriers [129,130]. These barriers play an important role in protecting against foreign antigens, microbes, and potentially harmful elements entering systemic circulation. Gut injury may lead to impaired gut barrier function, which may result in multiple organ dysfunction. Early in the 2000s, multiple studies drew a link between the gut and multiple organ dysfunction syndromes (MODS) and proposed that the gut acts as a motor of organ dysfunction [131,132].

### 6.1. Clinical Evidence

Recent clinical and experimental studies have shown communication between the kidney and gut, which has sparked new research and much debate concerning the pathophysiology and treatments. However, most studies have mainly tackled the development and progression of CKD or ESRD, and limited research is available regarding the relationship between AKI and gut dysbiosis [133,134]. Gut dysbiosis is the most common source of secondary infections in septic AKI patients, particularly those in ICU [135]. After gut injury, the intestinal epithelium permeability is increased, which results in the translocation of bacteria, toxins, and microbiota-derived metabolites from the intestinal lumen to the mesenteric lymph and systemic circulation, slowing renal recovery and increasing mortality.

Moreover, Nakade et al. have revealed that the level of serum D-serine significantly correlates with decreased kidney function in AKI patients [136]. A prospective cohort study showed that when compared with healthy subjects, serum indoxyl sulfate (IS) is significantly elevated in patients with hospital-acquired AKI and is associated with a poor prognosis [137]. Notably, accumulating clinical data about the immunomodulatory role of the gut microbiome in patients with CKD suggests the potential role of the intestinal microbiota in kidney–gut crosstalk in AKI patients [129,138], but more related clinical research is needed.

### 6.2. Laboratory Evidence

AKI-induced gastrointestinal homeostasis disruption has been confirmed in the laboratory. Recently, Yang et al. found that the gut microbiota *Escherichia coli* and *Enterobacter* were increased, but *Lactobacillus*, *Ruminococcaceae*, *Faecalibacterium*, and *Lachnospiraceae* were decreased in an IRI-AKI model. Moreover, the germ-free mice which received the post-AKI microbiota exhibited more serious kidney injury and inflammation after an ischemia-reperfusion operation compared with the mice which received the microbiota from a sham group [139]. On the other hand, some metabolic products of microbiota, such as short-chain fatty acids (SCFAs), have a protective effect on AKI. Administration of the three main SCFAs, acetate, propionate and butyrate, can improve kidney injury in AKI by decreasing local and systemic inflammation, oxidative stress, inflammatory cell infiltration, and cell apoptosis [140]. Zou et al. also found that the gut microbiota facilitates the therapeutic effect of Qiong-Yu-Gao, a traditional Chinese medicine, on cisplatin-induced AKI [141]. These results indicate the dual relationship between the intestine and kidneys and suggest that intestinal microbiota may be a potential target for AKI treatment.

### 6.3. Mechanism

#### 6.3.1. Intestinal Microbiota and Its Products

Increasingly, it is recognized that the intestinal microbiota plays a crucial role in the gut–kidney axis. In the process of AKI, the microbiota can produce compounds that have protective or harmful effects on the kidneys. SCFAs, including acetate, propionate, and butyrate, are one kind of nephroprotective compound produced from indigestible food in the colon by the gut microbiota [142]. The protective effects of SCFAs mainly rely on the activation of G-protein-coupled receptors (GPCRs), GPR41, GPR43, Olfr78, and GPR109, and the inhibition of histone deacetylase (HDAC) [143,144,145]. Both in vivo and in vitro studies have revealed that treatment with SCFAs not only decreases the production of reactive oxygen species (ROS), and cytokines, such as IL-1β, IL-6, TNF-α, and MCP-1, but also improves mitochondrial biogenesis in tubular epithelial cells. In addition, the activation of NF-κB signaling and the expression of TLR4 are inhibited [140]. Furthermore, in vitro research has shown that SCFAs inhibit dendritic cell maturation and block the capacity of these cells to induce CD4 and CD8 T cell proliferation [146,147]. These findings indicate that SCFAs may be a novel therapeutic strategy for AKI.

On the other hand, microbiota-derived uremic toxins can cause further kidney damage in AKI. Several uremic toxins have been identified, such as indoxyl sulfate (IS), para-cresyl sulfate (PCS), Trimethylamine-N-oxide (TMAO), indole-3 acetic acid (IAA), etc. [148,149]. Normally, circulatory uremic toxins are excreted from the kidney through organic anion transporters (OATs), but kidney dysfunction leads to their accumulation in the kidney. Accumulated uremic toxins are responsible for endothelial dysfunction, ROS production, pro-inflammatory factor expression, and leukocyte extravasation, eventually exacerbating kidney damage [150].

#### 6.3.2. Inflammation

During AKI, increased inflammatory cytokines and activated immune cells can cause gut barrier function damage and increase permeability. Intestinal hyper-permeability is a common reason for aggravating kidney failure in AKI patients. The intestinal epithelium apical tight junctions and junctional adhesion molecules (JAM) contribute to the integrated gut’s barrier function, which prevents the luminal contents from escaping into the internal environment. In septic AKI, the expression of zonulaoccludens-1 (ZO-1), one of the multiple claudin isoforms in the tight junction complex, is downregulated, leading to intestinal hyper-permeability [151]. In addition, the cytokines released from neutrophils or macrophages can activate myosin light chain kinase (MLCK) which phosphorylates the myosin light chain, causing the further opening of the tight apical junction [152].

#### 6.3.3. Urea Accumulation

Furthermore, urea accumulation caused by kidney dysfunction is another cause of increased intestinal permeability. Urea diffuses from the blood into the gut lumen, where it is converted to ammonia by gut bacterial urease (CO(NH_2_)_2_ + H_2_O → CO_2_ + 2NH_3_). The ammonia is transformed into caustic ammonium hydroxide (NH_3_ + H_2_O → NH_4_OH), which can destroy the tight junction proteins that keep epithelial cells together [135,153]. Hence, the toxins and bacteria in the intestinal lumen tend to transfer into the systemic circulation, which in turn triggers a more severe inflammatory response, further inducing kidney injury, organ failure, and higher mortality (Figure 4).

## 7. Kidney–Brain Crosstalk

The function of the kidney in controlling sodium and water reabsorption is largely regulated by the brain. Under physiological conditions, the neurons in the supraspinal nucleus and paraventricular nucleus of the hypothalamus secrete AVP, which acts on distal convoluted tubules and collects duct epithelial cells to promote the absorption of water [154]. In addition, renal blood flow and glomerular filtration rate are mediated by changes in the sympathetic nervous system via renal vasoconstriction and the RAAS [155]. These phenomena indicate the existence of kidney–brain crosstalk.

### 7.1. Clinical Evidence

AKI is a frequently encountered complication in patients with brain injury such as stroke. A meta-analysis of 12 studies, which included 4,532,181 acute ischemic stroke patients and 615,636 intracerebral hemorrhage patients, found that the pooled prevalence rate of AKI after all stroke types was 11.6%, and that AKI is associated with an increased mortality rate [156]. In a 10-year follow-up study of a large cohort of first-ever acute stroke patients (involving 2155 patients), approximately 27% of patients developed AKI after acute stroke, and the probability of 10-year mortality of patients with AKI was 75.9% [157]. Moreover, stroke patients with severe neurological deficits and cardiac abnormalities such as heart failure, atrial fibrillation, ischemic heart disease, hyperglycemia, hypertension, low eGFR, or advanced age were more susceptible to developing AKI [158].

In turn, encephalopathy and mental disorders including somnolence, epileptic, coma and chorea are common complications of AKI, which may occur simultaneously with AKI or develop subsequently [159]. Several clinical studies have identified the long-term neurologic deficits of AKI. A nationwide population study involving 4315 AKI-recovery patients and 4315 non-AKI control subjects revealed that the AKI-recovery patients had a higher risk, a higher severity of stroke events and, moreover, a higher risk of mortality than the non-AKI group, regardless of progression to subsequent chronic kidney disease [160]. This impact was similar to diabetes.

### 7.2. Laboratory Evidence

Several experimental studies have demonstrated the communication between the kidney and brain in animal models. Rabb et al. revealed that when compared with a sham group, a mouse renal IRI AKI model increased neuronal pyknosis and microgliosis in the brain and disrupted the blood-brain barrier. In addition, AKI led to the accumulation of proinflammatory chemokines, including keratinocyte-derived chemoattractant and G-CSF, in the cerebral cortex and hippocampus, and increased expression of glial fibrillary acidic protein in astrocytes in the cortex and corpus callosum [161]. In an SD rat IRI model, TLR4 in the hippocampus and striatum were significantly upregulated, suggesting that TLR4 plays a critical role in AKI-induced neuroinflammation [162]. Moreover, the turnover of dopamine in the striatum, mesencephalon, and hypothalamus was decreased in IRI rats, which impaired the motor activity of rats [163].

### 7.3. Mechanism

#### 7.3.1. Uremic Toxins

Patients with AKI are more susceptible to encephalopathy compared with patients with CKD, probably due to high uremic toxins [164,165]. In an IRI model, accumulated uremic toxins impaired motor activity and led to abnormal behavior in rats [163]. However, the mechanism of these pathological changes in the brain during AKI is still awaiting clarification.

#### 7.3.2. Inflammation

In addition, animal studies have identified that inflammation is activated in the brain following AKI, and these changes are accompanied by increased vascular permeability in the brain [161,164], suggesting a disruption of the blood-brain barrier (BBB). Injury to the kidney increases the release of cytokines and chemokines, such as TNF-α, IL-6, IFN-γ, MCP-1, and CXCL-1, which then infiltrate into the brain through the disrupted BBB, leading to brain injury [166,167]. On the other hand, the injured brain releases the pro-inflammatory molecules into the blood which then promote the infiltration of macrophages and neutrophils into the kidney. Notably, the recruitment of inflammatory cells is a hallmark of AKI pathogenesis and an important cause of renal tubular cell apoptosis [107].

#### 7.3.3. Exosomes

Exosomes are endogenously produced, membrane-bound vesicles that contain various molecules, including mRNA, miRNA, lncRNA, and small molecular proteins. Moreover, exosomes have been implicated in the pathogenesis of AKI, CKD, renal fibrosis, end-stage renal disease (ESRD), glomerular diseases, and diabetic nephropathy [168]. One basic research study demonstrated that AKI and CKD mice secreted high levels of exosomes from kidneys and in urine, which contained elevated levels of inflammatory cytokine mRNA when compared with normal mice [166,169], suggesting that exosomes may participate in cell or organ communication. However, further studies are needed in this area.

## 8. Therapeutic Strategy and Management

Depending on its severity, the long-term outcomes of AKI include GFR decreases, nephron loss, and increased risk of CKD, cardiovascular disease, and kidney cancer [2,170,171]. When AKI follows other organ failures, thanks to the potential self-repairing capability of the kidney, the symptoms of AKI can recover after removal of the pathogeny in other organs (within 24–48 h) [172]. If AKI persists for ≥72 h, these patients have considerably worse outcomes [173].

The concept of kidney lifespan can be used to indicate the prognosis of AKI. The number of healthy nephrons determines the individual kidney lifespan. Notably, the number of nephrons is set at birth and starts to decline with age around the age of 25 [174]. By age 70, healthy individuals have only half of original nephron number left [175]. AKI can cause irreversible loss of nephrons. Once the number of nephrons drops to a certain level, kidney lifespan shortens and renal dysfunction occurs [2,176]. Hence, for patients with a high nephron endowment or those at a young age, there may be no consequence after AKI. Patients with fewer nephrons or at an older age may have a poor outcome.

Suitable therapeutic strategy and management probably improve the outcome of AKI. However, no specific drug has been consistently and reproducibly shown to have a protective effect on the kidneys. In the setting of AKI-induced distant dysfunction, anti-oxidants, anti-inflammatory agents, and renal replacement therapy are consequentially helpful based on the mechanisms discussed above.

### 8.1. Therapeutic Strategy

#### 8.1.1. Anti-Oxidants

Several groups have revealed that the administration of anti-oxidants, such as prostaglandin E1 (PGE1), glutathione, vitamin E, and thymoquinone, significantly alleviates lung and liver injury after AKI [117,177,178,179]. In addition, improving the activation of endogenous anti-oxidant factors, including superoxide dismutase (SOD), heme oxygenase 1 (HO-1), glutathione-S-transferase (GST), and glutathione peroxidase (GPX), is considered a promising therapeutic strategy in AKI-induced distant dysfunction [180,181]. The critical role of these anti-oxidative enzymes in organic injury has been widely studied and summarized in other reviews.

#### 8.1.2. Anti-Inflammatory Agents

Severe inflammation mediates the damage to organs induced by AKI. Hence, anti-inflammation is one of the main therapeutic approaches to treat organ injury. Delivering IL-10, or blocking IL-6, CXCL1, and TNF-α have been demonstrated to alleviate AKI-induced lung injury [182,183,184]. Moreover, NF-κB signaling activated by DAMP ligation of pattern recognition receptors (PRRs) also results in pro-inflammatory cytokines and perpetuating the inflammatory response, making TLR4 another attractive therapeutic target for distant organs dysfunction induced by AKI [185].

#### 8.1.3. Renal Replacement Therapy

Renal replacement therapy (hemodialysis) is commonly applied to provide renal support for critically ill patients with AKI, especially those with hemodynamic instability [186]. Renal replacement therapy can aid in the removal of metabolic acidosis, electrolyte disturbances, volume overload, and uremic toxins. Thus, renal replacement therapy has many theoretical advantages in relieving the injury of distant organs caused by AKI [187]. Indeed, renal replacement therapy may enable support of not only the kidney, but also the heart, the lung, and the liver. However, the optimal time to start renal replacement therapy in patients with AKI remains controversial, especially in specific populations [188]. Therefore, its effect should be further evaluated in the future.

#### 8.1.4. Other Therapies

Other therapeutic strategies including interventions with RAAS inhibitors, atrial natriuretic peptide (ANP), probiotics, and prebiotics, could be promising approaches targeting the crosstalk between injured kidneys and other organs.

### 8.2. Management

Considering AKI is not a disease but rather a loose collection of syndromes, the first step in managing AKI is to determine its cause [2]. However, the importance of AKI management is often overlooked. An audit in the UK showed that more than 50% of patients with AKI were managed badly, and 43% of AKI cases were recognized late or not at all [189]. Notably, a clinical observation including 1893 patients undergoing partial nephrectomy found that 20% of patients experienced AKI after surgery. The rate of patients recovering 90% of baseline function was lower, and the CKD upstaging proportion was higher in the AKI group compared with those without AKI. Moreover, longer AKI (≥4 days) increases the risk of functional deterioration and CKD development [190], suggesting early intervention in AKI is critical. The establishment of fully equipped AKI management is an important measure to prevent the development of AKI-induced distal organ dysfunction into multiple organ failure (MOF).

#### 8.2.1. Volume Status and Hemodynamic Management

Volume status and hemodynamics should be carefully reviewed in patients with severe AKI. When the renal hypoperfusion (e.g., cardiac dysfunction) is sustained and the adaptive response is inadequate, renal damage can occur due to inadequate oxygen and nutrient delivery, and epithelial cell injury or death [191,192]. On the other hand, patients with AKI can suffer fluid overload due to reductions in urine output, which can directly damage the kidney parenchyma and have considerably harmful effects on other organs (e.g., pulmonary edema) [2].

In physiological conditions, the kidneys and other organs are adequately perfused at a mean arterial pressure of 65 mmHg [193]. Normal arterial pressure is critical for maintaining the blood perfusion of the kidneys. It is notable that hypotension frequently indicates kidney hypoperfusion despite apparent hypervolaemia, in which fluid redistributes to the venous system or into tissue interstitium (e.g., congestive heart failure, capillary leakage during systemic inflammatory response syndrome (SRIS)) [2,7]. Thus, an increase in mean arterial pressure should be considered in some cases.

#### 8.2.2. Nephrotoxic Agent and Drugs Management

Clinically, the phenomenon that drugs aggravate AKI should not be ignored. All potential nephrotoxic agents that can be stopped should be avoided. Indispensable drugs should only be used at the required dose for the time needed and carefully monitored. If possible, drug concentration monitoring should be established [194]. Contrast or radiocontrast agents (e.g., iodinated contrast medium, paramagnetic ion complexes, or superparamagnetic magnetite particles) are widely used in medical imaging [195]. Excessive exposure to contrast agent will directly cause tubular injury and renal hypoperfusion [196]; therefore, their use should be limited in patients with severe AKI. Although contrast-associated AKI is becoming less common thanks to reduced toxicity and lower amounts of contrast media used for imaging techniques [197,198], their potential nephrotoxicity cannot be ignored.

#### 8.2.3. Additional Notes

Blood electrolyte solution imbalances should be noted, especially in critically ill patients. Non-physiologic ratios of sodium and chloride may worsen AKI [199]. In addition, a cohort study including 1132 patients with urinary tract infections found that about 14% of these patients developed AKI, suggesting urinary tract infection may be a potential factor for aggravating AKI [200]. Obstructive nephropathy or nephrolithiasis is also a common cause of AKI, especially in young children (up to 30%) [201]. Thus, for these patients, the first step should be to remove the cause of AKI, and if necessary, clinicians must personalize the care of patients. Considering renal scintigraphy is a powerful imaging method with unique advantages for assessing renal function [202,203] independent of changes in plasma creatinine and fluid volume, it is potentially useful in the management of AKI patients.

## 9. Conclusions

AKI is not only a common complication following other organ failures, but also causes distant organ dysfunction and is an important cause of high mortality in hospitalized patients. In this review, we examined the clinical and laboratory evidence concerning organ crosstalk in AKI, and summarized the underlying mechanisms of AKI-induced distant dysfunction and organ failure-induced AKI based on the current findings. These mechanisms include inflammation, immune responses, hemodynamic change, fluid homeostasis, hormone secretion, nerve reflex regulation, uremic toxins and oxidative stress. Besides these factors, exosomes have been demonstrated to participate in mediating muscle–kidney [204], adipose tissue–pancreas, retina, hippocampus, and hypothalamus crosstalk [205], but evidence for the potential role of exosomes in AKI is still sparse and should be a particular focus of future research. Understanding the complex interactions between kidneys and other organs may help to develop new diagnostic approaches and therapeutic strategies to improve outcomes in patients with AKI.

## Figures and Tables

**Figure 1 jcm-11-06637-f001:**
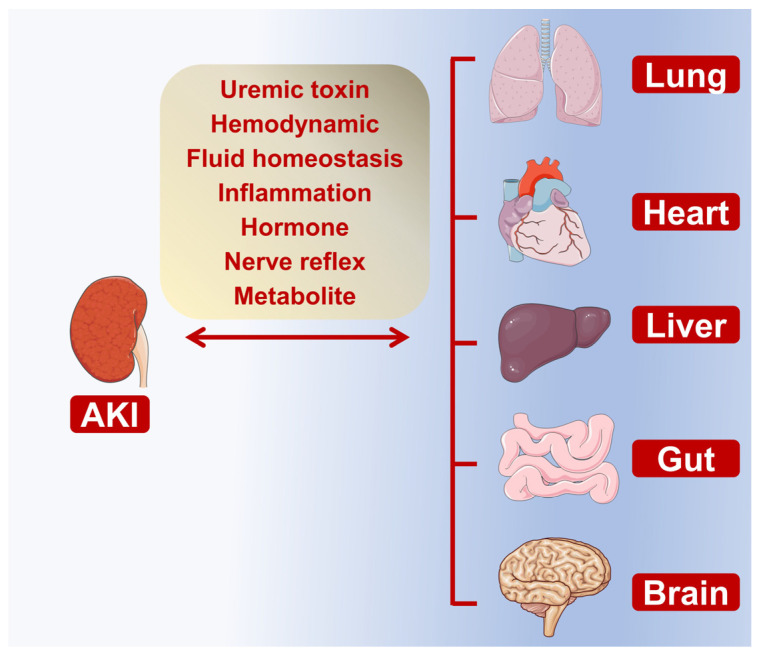
Overview of organ crosstalk during AKI. The crosstalk between kidney and other organs, including lung, heart, liver, gut, and brain, under AKI conditions. The underlying mechanism involves uremic toxin retention, hemodynamic change, fluid homeostasis damage, neuro-hormonal dysfunction, inflammation, and metabolite accumulation. Abbreviations: AKI, acute kidney injury.

**Figure 2 jcm-11-06637-f002:**
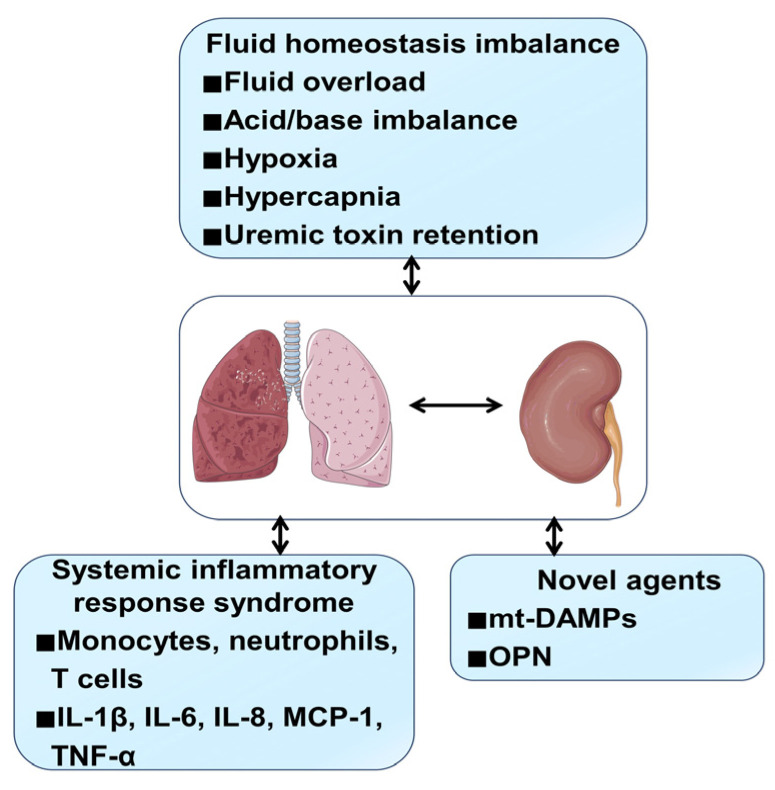
Mechanisms of kidney–lung interactions. Fluid homeostasis imbalance, systemic inflammatory response syndrome, and some novel agents contribute to AKI-induced acute lung injury (ALI) or ARSD-induced kidney injury. The main fluid homeostasis imbalance mechanisms include fluid overload, acid/base imbalance, uremic toxin retention, hypoxia, and hypercapnia. Systemic inflammatory response syndrome mechanisms include pro-inflammatory factors release and immune cells response. Moreover, some novel mediators, such as mt-DAMPs and OPN, were identified as participating in kidney–lung interactions. Abbreviations: ARSD, acute respiratory distress syndrome; mt-DAMPs, mitochondrial damage associated molecular patterns; OPN, osteopontin.

**Figure 3 jcm-11-06637-f003:**
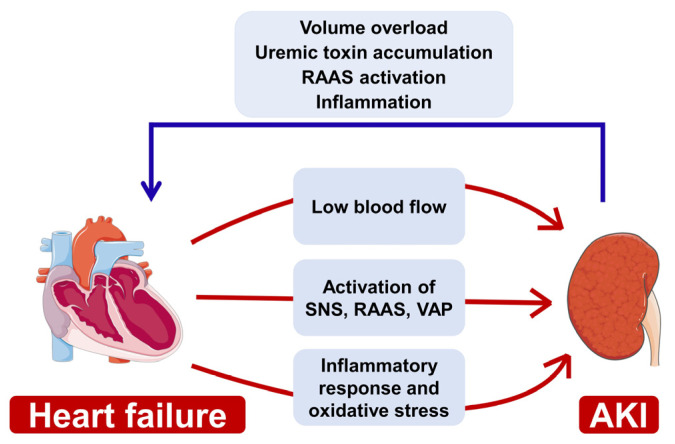
Overview of key kidney–heart interactions. Three major mechanisms contribute to the development or acceleration of cardio-renal dysfunction, including low renal afferent blood flow, activation of SNS, RAAS, VAP, systematic inflammation, and oxidative stress. On the other hand, AKI-induced volume overload, uremic toxin retention, and RAAS over-activation accelerate heart failure. Abbreviations: SNS, sympathetic nervous system; RAAS, renin–angiotensin–aldosterone system; VAP vasopressin.

**Figure 4 jcm-11-06637-f004:**
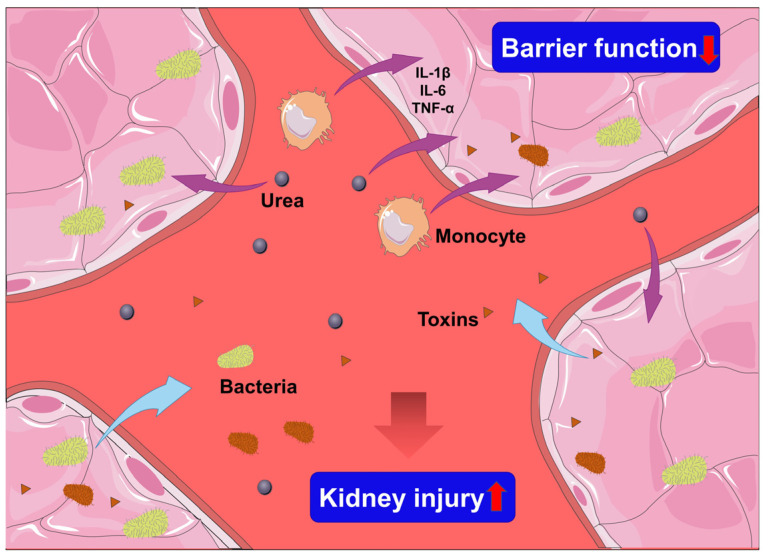
Mechanisms of kidney–gut interactions. During AKI, the accumulation of urea and the infiltration of monocytes cause gut barrier damage, which then allows harmful bacteria and their toxic metabolites (toxins) to leak into the circulation system and leads to further kidney injury. The brown and yellow bateria represent different species of bateria in the gut. The purple arrow indicate the transfer of urea or inflammatory cell form blood into gut. The bule arrow indicate bateria or toxins entering circulation form gut.

**Table 1 jcm-11-06637-t001:** Classification and criteria of AKI, AKD, and CKD.

	Duration	Functional Criteria
AKI	≤7 days	
Subclinical AKI		TIMP-2*IGFBP-7 > 0.3 (ng/mL)^2^/1000
AKI stage 1		Increase in scr by 1.5–1.9 times baseline within 7 d or increase in scr by ≥0.3 mg/dL in 48 h or UO < 0.5 mL/kg/h for 6 h
AKI stage 2		Increase in scr by 2–2.9 times baseline within 7 d or UO < 0.5 mL/kg/h for 12 h
AKI stage 3		Increase in scr by 3 times baseline within 7 d or increase in scr by ≥4 mg/dL or UO < 0.5 mL/kg/h for 24 h or anuria for ≥12 h
AKD	<3 months	AKI or GFR < 60 mL/min/1.73 m^2^ or decrease in GFR by ≥35% over baseline or increase in scr by >50% over baseline
CKD	>3 months	GFR < 60 mL/min/1.73 m^2^

Abbreviations: AKI, acute kidney injury; AKD, acute kidney diseases and disorders; CKD, chronic kidney disease; scr, serum creatinine; UO, urine output; GFR, glomerular filtration rate.

**Table 2 jcm-11-06637-t002:** Classification and definition of cardiorenal syndrome based on the ADQI.

Classification	Definition	Clinical Examples
CRS type 1 (Acute cardiorenal syndrome)	Acute worsening of heart function resulting in kidney injury and/or dysfunction	ACS, AHF, and cardiogenic shock leading to AKI
CRS type 2 (Chronic cardiorenal syndrome)	Chronic heart failure resulting in kidney injury or dysfunction	CHD such as congestive heart failure and cardiomyopathy leading to CKD
CRS type 3 (Acute reno-cardiac syndrome)	Acute worsening of kidney function resulting in heart injury and/or dysfunction	AHF, ACS, uremic cardiomyopathy, and arrhythmias secondary to AKI
CRS type 4 (Chronic reno-cardiac syndrome)	CKD resulting in heart injury, disease, and/or dysfunction	AHF, ACS, and CHD secondary to CKD
CRS type 5 (Secondary cardio-renal syndrome)	Systemic disorders resulting in simultaneous injury and/or dysfunction of heart and kidney	Vasculitis, sepsis, and cirrhosis causing AHF, ACS, CHD, AKI, and CKD

Abbreviations: ADQI, acute dialysis quality initiative; CRS, cardiorenal syndrome; ACS, acute coronary syndrome; AHF, acute heart failure; AKI, acute kidney injury; CKD, chronic kidney disease; CHD, chronic heart disease.

## Data Availability

Not applicable.

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
