# Peer review of "Organ Crosstalk in Acute Kidney Injury: Evidence and Mechanisms"

_jcm, 2022, doi:10.3390/jcm11226637_

Round 1

Reviewer 1 Report

This is a very extensive narrative review article which has been well structured. I would suggest more time spent on the initial definition of AKI would be valuable to the reader, as well as the classification of AKI with respect to Pre-renal , Renal and Post Renal. Very little of the manuscript is devoted to therapeutic intervention and would suggest they address this imbalance. Minor English language errors throughout, but well written apart from suggesting that AKI resolves, as if without any long term sequelae.    

Author Response

Response to reviewer 1:

This is a very extensive narrative review article which has been well structured. I would suggest more time spent on the initial definition of AKI would be valuable to the reader, as well as the classification of AKI with respect to Pre-renal, Renal and Post Renal. Very little of the manuscript is devoted to therapeutic intervention and would suggest they address this imbalance. Minor English language errors throughout, but well written apart from suggesting that AKI resolves, as if without any long term sequelae.

Reply: Thank you for your comments. We have added the definition and classification of AKI in the introduction and Table 1. In the part of AKI therapy (Part 7), we have added the contents associated with the management and long-term outcome of AKI. We have corrected English language errors in the full text.

Reviewer 2 Report

Important revisions

The work is well constructed and summarizes the impact of AKI on the various organs and apparatus and follows in the footsteps of Kellum's work of 2021.

1) Add a chapter dedicated exclusively to AKI and the vascular system already definition cited as circulatory dysfunctions. It is necessary to comment on the impact of AKI on changes in blood pressure, state of dehydration and hyperhydration, hypervolemia and hypovolemia. I would mention the most toxic molecules to the endothelial system. Add an image of the arterial and venous system to figure 1;

2) it is necessary to comment on the average age of the patients in the various severity ranges of AKI;

3) In the inter-relationship between AKI and the liver, add the hepatorenal syndrome with causes, mechanisms. This is must be consistently integrated.

4) Carry out some updates on laboratory tests taking into account, for example, Husain-Syed's latest work of 2022.

4) it is useful to dedicate a few lines for each AKI-organ chapter about a warning between AKI and drugs, citing the main causal or worsening factors drugs in order to make the work more usable as a “take home message”.

Minor revisions

It is not clear the phrase "is limited 7 days". Explain and specify better perhaps referring to a bibliographic entry.

Author Response

Response to reviewer 2:

Important revisions

The work is well constructed and summarizes the impact of AKI on the various organs and apparatus and follows in the footsteps of Kellum's work of 2021.

1) Add a chapter dedicated exclusively to AKI and the vascular system already definition cited as circulatory dysfunctions. It is necessary to comment on the impact of AKI on changes in blood pressure, state of dehydration and hyperhydration, hypervolemia and hypovolemia. I would mention the most toxic molecules to the endothelial system. Add an image of the arterial and venous system to figure 1;

Reply: Thank you for your insightful comments. We think it would certainly improve the quality of our manuscript. We have added a narration of studies in which vascular dysfunction promotes AKI (Parts 3) and add a related image in Figure 1. The comment on the impact of AKI on changes in blood pressure, state of dehydration and hyperhydration, hypervolemia and hypovolemia were shown in parts 3 and AKI management (Parts 7).

2) it is necessary to comment on the average age of the patients in the various severity ranges of AKI;

Reply: Thank you for your comments. We have added the comments on the influence of age on the prognosis of AKI in part 7, AKI therapy strategy.

3) In the inter-relationship between AKI and the liver, add the hepatorenal syndrome with causes, mechanisms. This is must be consistently integrated.

Reply: Thank you for your comments. We have described the causes and mechanisms of hepatorenal syndrome in the original text (Part 4, 4.3 mechanism), and we also have noticed liver-kidney communication in addition to it.

4) Carry out some updates on laboratory tests taking into account, for example, Husain-Syed's latest work of 2022.

Reply: Thank you for your comments. We have updated some laboratory tests. Based on Husain-Syed's latest work, we have added the concept of subclinical AKI and referred to biomarkers that are currently used clinically to detect early AKI in the first paragraph.

5) it is useful to dedicate a few lines for each AKI-organ chapter about a warning between AKI and drugs, citing the main causal or worsening factors drugs in order to make the work more usable as a “take home message”.

Reply: Thank you for your comments. In part 7 (7.2.2 Nephrotoxic drugs management), we have added the comments on the management of drugs in AKI patients and the precautions of the use of potentially nephrotoxic agents.

Minor revisions

It is not clear the phrase "is limited 7 days". Explain and specify better perhaps referring to a bibliographic entry.

Reply: Thank you for your comments. We have corrected English language errors in this part.

Reviewer 3 Report

The review by Li and colleagues is relevant from both, a clinical and a scientific perspective. It illuminates cellular / molecular mechanisms involved in crosstalk between kidney and various other organs. The poor prognosis of AKI subjects is hardly explainable by the transient or prolonged loss of excretory function alone, since kidney replacement therapy allows to eliminate most deleterious substances usually eliminated in the urine. Therefore, numerous additional mechanisms must necessarily account for generalized organ dysfunction in AKI. The authors put great effort in summarizing the literature on the topic. The article is worth to be published. However, the language needs to be polished, for instance:

line 39: However, there are exist a significant…

line 52: However, thanks to the limit …

line 75: he lung and kidney sophisticatedly cooperate…

and others

Author Response

Response to reviewer 3:

The review by Li and colleagues is relevant from both, a clinical and a scientific perspective. It illuminates cellular / molecular mechanisms involved in crosstalk between kidney and various other organs. The poor prognosis of AKI subjects is hardly explainable by the transient or prolonged loss of excretory function alone, since kidney replacement therapy allows to eliminate most deleterious substances usually eliminated in the urine. Therefore, numerous additional mechanisms must necessarily account for generalized organ dysfunction in AKI. The authors put great effort in summarizing the literature on the topic. The article is worth to be published. However, the language needs to be polished, for instance:

line 39: However, there are exist a significant…

line 52: However, thanks to the limit …

line 75: he lung and kidney sophisticatedly cooperate…

and others

Reply: Thank you for your insightful comments. We have corrected English language errors in the full text.

Reviewer 4 Report

Add methodological Issues:

The kind of review you wrote and search criterion

Add paragraphes including ischemia role especially in post surgical AKI with relevant literature not considered ( eg surgical vs medical AKi; nephron sparing surgery etc)

Note that clinical scenarios you included focusing on two organs cross talks very often can be gathered as MOF in critical patients. ( It could be appreciable if you develope this issue as well)

A paragraph or discussion on AKI and urinary infections should be added ( eg: obstructive nephropathy and pielonephritis); very often the two clinical scenarios walk togheter determining critical mulit-organs crosstalks and impairment in SIRS and Sepsis.

Author Response

Response to reviewer 4:

Add methodological Issues: The kind of review you wrote and search criterion

Reply: Thank you for your comments. This is an inductive review in which we mainly searched the most relevant literatures on each topic from PubMed.

Add paragraphs including ischemia role especially in post surgical AKI with relevant literature not considered ( eg surgical vs medical AKi; nephron sparing surgery etc)

Reply: Thank you for your insightful comments. We think it would certainly improve the quality of our manuscript.  We have added the narration of the role of ischemia in AKI development in parts 3 (3.3.1 Ischemia and low blood flow) and in parts 7 (7.2.1 Volume status and hemodynamic management).

Note that clinical scenarios you included focusing on two organs cross talks very often can be gathered as MOF in critical patients. ( It could be appreciable if you develope this issue as well)

A paragraph or discussion on AKI and urinary infections should be added ( eg: obstructive nephropathy and pielonephritis); very often the two clinical scenarios walk togheter determining critical mulit-organs crosstalks and impairment in SIRS and Sepsis.

Reply: Thank you for your insightful comments. We think it would certainly improve the quality of our manuscript. We have added the new comments on MOF and urinary infections in the part of AKI management (7.2 Management).

Round 2

Reviewer 2 Report

No comments

Author Response

Reply: Thank you for your appreciation!

Reviewer 4 Report

Thank you for adding relevant topics on ischaemia role and multiorgan cross-talk

some topic as

-Surgical induced AKI after Partial nephrectomy,

-Contrast media effect (both iodine and paramagnetic)

-Role of renal scintigraphy

can be mentioned for fully cover the very relevant topic you treated with a very well planned work

suggestions:

  Eur Urol. 2019 Sep;76(3):398-403. doi: 10.1016/j.eururo.2019.04.040. Epub 2019 May 10. Impact of Acute Kidney Injury and Its Duration on Long-term Renal Function After Partial Nephrectomy

Carlo Andrea Bravi et Al 

Urologia. 2009 Jan-Mar;76(1):10-8.Contrast media in urogenital radiology. N Foschi et Al

BJU Int. 2015 Apr;115(4):606-12. Evaluation of functional outcomes after laparoscopic partial nephrectomy using renal scintigraphy: clamped vs clampless technique. Porpiglia F et Al 

Nucl Med Commun. 2021 Jun 1;42(6):602-610. Advantages of gravity-assisted diuretic renogram: F + 10 (seated position) method. Girolamo Tartaglione et al 

Author Response

Reply: Thank you for your comments and helpful suggestions. We have added the comment on contrast agent application, partial nephrectomy, and renal scintigraphy in part 7 (7.2 Management).
